# Study on Structure Activity Relationship of Natural Flavonoids against Thrombin by Molecular Docking Virtual Screening Combined with Activity Evaluation In Vitro

**DOI:** 10.3390/molecules25020422

**Published:** 2020-01-20

**Authors:** Xiaoyan Wang, Zhen Yang, Feifei Su, Jin Li, Evans Owusu Boadi, Yan-xu Chang, Hui Wang

**Affiliations:** 1Tianjin State Key Laboratory of Modern Chinese Medicine, Tianjin University of Traditional Chinese Medicine, Tianjin 300193, China; wang_xiaoyan27@163.com (X.W.); zishan826@126.com (J.L.); kwakuboadi17@yahoo.com (E.O.B.); 2Tianjin Key Laboratory of Phytochemistry and Pharmaceutical Analysis, Tianjin University of Traditional Chinese Medicine, Tianjin 300193, China; 3Tianjin Key Laboratory of Chinese Medicine Pharmacology, Tianjin University of Traditional Chinese Medicine, Tianjin 300193, China; yzwygb@126.com (Z.Y.); sufeifeilch@163.com (F.S.); 4College of Chinese Materia Medica, Tianjin University of Traditional Chinese Medicine, Tianjin 300193, China

**Keywords:** flavonoids, molecular docking, structure activity relationship, thrombin

## Abstract

Thrombin, a key enzyme of the serine protease superfamily, plays an integral role in the blood coagulation cascade and thrombotic diseases. In view of this, it is worthwhile to establish a method to screen thrombin inhibitors (such as natural flavonoid-type inhibitors) as well as investigate their structure activity relationships. Virtual screening using molecular docking technique was used to screen 103 flavonoids. Out of this number, 42 target compounds were selected, and their inhibitory effects on thrombin assayed by chromogenic substrate method. The results indicated that the carbon-carbon double bond group at the C2, C3 sites and the carbonyl group at the C4 sites of flavones were essential for thrombin inhibition, whereas the methoxy and O-glycosyl groups reduced thrombin inhibition. Noteworthy, introduction of OH groups at different positions on flavonoids either decreased or increased anti-thrombin potential. Myricetin exhibited the highest inhibitory potential against thrombin with an IC_50_ value of 56 μM. Purposively, the established molecular docking virtual screening method is not limited to exploring flavonoid structure activity relationships to anti-thrombin activity but also usefully discovering other natural active constituents.

## 1. Introduction

In recent years, thrombotic disease has been a major threat to human health worldwide. As mentioned earlier, thrombin, a principal serine protease, plays a critical role in blood coagulation cascade response by activating blood platelet and catalyzing the conversion of soluble plasma-protein fibrinogen to insoluble fibrin [1,2]. This implies that thrombin metabolic disorder is a major triggering factor for thrombotic diseases. In clinical practice, there are some effective anti-thrombin inhibitors such as bivalirudin, argatroban, heparin among others. However, the cost of treatment with these drugs is expensive and also associated with serious side effects [2,3,4]. These reasons underscore the need to seek and discover safer, affordable and effective natural thrombin inhibitors for the treatment of thrombotic disease.

In recent research, plant-derived flavonoids, namely, kaempferol, quercetin and its glycosylated derivatives, (+)-catechin, (-)-epicatechin, cyanidin, cyanidin-3-glucoside, eupatilin-7-*O*-β-d-glucopyranoside and 5,6,2′,4′-tetrahydroxy-7,5′-dimethoxyflavone have been proven to have inhibitory effect on thrombin [5,6,7,8]. Notably, Traditional Chinese Medicines (TCMs) such as *Typha* pollen, *Salvia miltiorrhizae*, Radix et rhizoma, Chuanxiong rhizoma and *Platycladi* cacumen are enriched with natural flavonoids and have been used for treating thrombotic diseases for centuries [9,10]. 

Structurally, 2-phenylchromanone and C6-C3-C6 form the backbone of flavonoids. Some well-known flavonoids include but are not limited to flavonols, isoflavones, flavones, flavanonols, anthocyanidins, chalcones, dihydrochalcone and aurones. [11,12]. These compounds elicit diverse pharmacological activity such as antioxidant, anti-inflammation, antibacterial, antithrombus and antineoplastic due to the difference in structural diversities in basic parent structure or substituent group [13]. Structure activity relationship (SAR) of flavonoids based on the above biological activities has gained lots of attention in recent years [14,15,16]. To the best of our knowledge, there is only one literature report on the SAR of flavonoids inhibiting thrombin [17]. Noticeably, it did not investigate anti-thrombin activity for all the substituent groups on flavonoid. In view of this, it was imperative to study the structural characteristics of more natural flavonoids for unraveling the mechanism of thrombin inhibition. 

Molecular docking technology, an algorithm that predicts the putative geometry of a protein–ligand complex has been successfully applied in the studies of SAR of compounds and TCM as well as predicting binding affinities of ligands [18,19,20,21,22]. Irrespective of these gains, there also exist disadvantages such as low accuracy and high false positive rate. A chromogenic substrate assay (CSA), which is a rapid and very effective technique is another in vitro method for examining thrombin inhibition activity [4,19,23]. This assay determines the activity of enzymes in samples by measuring the absorbance of colored p-nitroaniline produced by the reaction of blood coagulation factor (such as FX, thrombin and prothrombin) and chromogenic substrate [24]. Clinically, CSA is widely used in diagnosing and monitoring of hemophilia A/B, investigating anticoagulant effects and screening enzyme inhibitors [5,25,26]. Therefore, the combination of the chromogenic substrate method and molecular docking could greatly improve the screening accuracy. In this study, the SAR of 103 flavonoids was studied using molecular docking combined with chromogenic substrate assay. The findings of this research aim to provide a basis for the study of mechanism of thrombin inhibition and also serve as a model for studying new flavonoid-type thrombin inhibitors.

## 2. Results and Discussion 

### 2.1. The Feasibility of the Established Docking Protocol

To verify the feasibility of the established docking protocol, the original ligand molecule was extracted from the co-crystallized thrombin complex (2GDE and the original ligand SN3) and re-docked into 2GDE using CDOCKER. The conformation of the re-docked co-crystallized thrombin complex (2GDE and SN3) is shown in Figure 1. The value of low root mean-square deviation (RMSD) between re-docked conformation and the crystal conformation was 0.7457 Å. In general, if RMSD ≤ 2.0 Å, the molecular docking model should be considered as feasible [27]. 

### 2.2. In Silico Screening for Thrombin Inhibitors

One hundred and three flavonoids compounds were docked into the active site of thrombin model and identified by CDOCKER module in discovery studio 4.5 software (Accelrys, Biovia, Boston, MA, USA). The results showed 62 hit compounds with -CDOCKER_energy above 20.0 kcal/mol and 54 hit compounds with -CDOCKER_INTERACTION energy above 40.0 kcal/mol (see Appendix A). The docking effect was good since the -CDOCKER_energy was higher than 15 kcal/mol [24]. 

Based on the above results, the anti-thrombin activities of 42 target flavonoids were selected for further research using a chromogenic substrate in vitro method. Selection of flavonoids was done according to three principles. The first rule was the value of -CDOCKER_energy should exceed 25 kcal/mol or -CDOCKER_INTERACTION energy should exceed 45 kcal/mol in order to improve the accuracy of measurement results. Secondly, flavonoids including flavonols, flavones, flavanonols and catechins with comparable structural features were given preference. Lastly, naturally derived and easily obtained compounds were chosen. The chemical structures of the 42 flavonoids are shown in Appendix A.

### 2.3. In Vitro Screening for Thrombin Inhibitors

IC_50_ values and docking values -CDOCKER_energy and -CDOCKER_INTERACTION energy of the evaluated flavonoids are listed in Table 1. At 4 mg/mL substrate concentration, 22 flavones inhibited thrombin with IC_50_ values less than 500 μM. The IC_50_ value of the positive control, Argatroban was 1.86 μM whereas the test flavonoids, myricetin, scutellarein, isorhamnetin, myricitrin, baicalin, apigenin and hydroxygenkwanin were 56.5 μM, 70.8 μM, 72.2 μM, 79.5 μM, 88.6 μM, 96.2 μM and 99.7 μM, respectively. These flavonoids were classified as strong inhibitors. Moreover, compounds with IC_50_ value ranging from 100 μM to 200 μM were considered as moderate inhibitors. Kaempferol (107 μM), hispidulin (126 μM), diosmetin (131 μM), herbacetin (133 μM), luteolin (146 μM), luteoloside (155 μM), galangin (159 μM) and fisetin (178 μM) were considered as moderate inhibitors. Compounds such as quercetin, genkwanin, astragaline, morin, baicalein, rutin and naringin were classified as weak thrombin inhibitors since their IC_50_ values ranged from 200 μM to 500 μM.

Pearson correlation analysis was established using SPSS 21.0 (SPSS, Chicago, IL, USA) based on the anti-thrombin IC_50_, -CDOCKER_energy and -CDOCKER_INTERACTION energy of the 22 flavonoids with IC_50_ values less than 500μM. Results showed that IC_50_ was negatively correlated with -CDOCKER_energy (p(corr) = −0.55, *p* = 0.008) and positively correlated with -CDOCKER_INTERACTION energy (p(corr) = 0.45, *p* = 0.035). It was demonstrated that the target compound shows high anti-thrombin activity when the -CDOCKER_energy was higher and -CDOCKER_INTERACTION energy lower.

### 2.4. Binding Site of Myricetin, Scutellarein, Isorhamnetin and Myricitrin in Thrombin Model

Binding mechanisms of flavonoids and thrombin of strong inhibitors and moderate inhibitors were investigated using Discovery Studio (DS). The theoretical binding sites of these compounds interaction with thrombin model (2GDE) are shown in detail (Table 2). 

For a detailed observation, the top four flavonoids with the IC50 values less than 100 μM (myricetin, scutellarein, isorhamnetin and myricitrin) were selected. The binding conformations of the 4 target flavonoids at the active site of thrombin are shown in Figure 2. The OHs on sites, C5 and C7 of the A ring of myricetin interacted with related residues of 2GDE, which displayed the two OHs on these sites were beneficial for anti-thrombin activity. Furthermore, there were interactions between the OH groups at C3′, C4′ and 5′ sites at the C ring of myricetin and SER195, CYS191 and GLY216, respectively. It was worth noting that more OH groups on C ring could increase thrombin inhibition. With respect to scutellarein, the oxygen atom on its C1 site and hydroxyl on the B ring interacted with 2GDE. The steric hindrance effect was weak due to lack of substituent group and this provided an excellent opportunity for the whole flavonoid molecule to enter the interior of the protein and interact with the protein residues. For isorhamnetin, hydroxyls on the C5, C7 sites and O-methylation on the B ring interacted with 2GDE. The results showed that the presence of OCH_3_ on B ring improved bioactivity. Lastly, 2GDE residues of thrombin were bound by hydroxyl groups at C7 and C3′ sites and O-glycosyl groups on C3 site of myricitrin. The introduction of O-glycosyl group at the C3 site increased the interaction between the glycoside and protein; however, this hindered some interactions between other sites and the target.

### 2.5. The Structure–Activity Relationship (SAR) of Flavonoids on Inhibition of Thrombin 

Based on IC_50_ values and –CDOCKER_ and –CDOCKER_INTERACTION energies, the potential SARs of these structurally diverse flavonoids on the inhibitory effect of thrombin were analyzed as follows:

#### 2.5.1. The Influence of Double Bond, Carbonyl, Hydroxyl and O-glycosyl Groups at the C2, C3, C4 Sites of C Ring 

The presence of C2=C3 and C4=O bonds of flavones was essential for inhibition of thrombin. Lack of either led to the loss of thrombin inhibition, which was consistent with the published report [17]. For instance, quercetin (C2=C3, C4=O, IC_50_=205 μM) vs. dihydroquercetin (C4=O, without C2=C3, IC_50_ > 500 μM) and epicatechin (without C2=C3 and C4=O, IC_50_ > 500 μM). Myricetin (C2=C3, C4=O, IC_50_=56. 5 μM) vs. dihydromyricetin (C4=O, without C2=C3, IC_50_ > 500 μM) and (-)-epigallocatechin (without C2=C3 and C4=O, IC_50_ > 500 μM). Apigenin (C2=C3, C4=O, IC_50_=96.2 μM) vs. naringenin (C4=O, without C2=C3, IC_50_ > 500 μM). Diosmetin (C2=C3, C4=O, IC_50_=131 μM) vs. hesperetin (C4=O, without C2=C3, IC_50_ > 500 μM). 

Moving forward, flavonoids with single hydroxyl group at C3 site were observed to have slightly decreased inhibitory activity. For example, apigenin (without C3-OH, IC_50_ = 96.3 μM) vs. kaempferol (with C3-OH, IC_50_ = 107 μM). Luteolin (without C3-OH, IC_50_ = 146 μM) vs. quercetin (with C3-OH, IC_50_ = 205 μM)). Notwithstanding the above, there was an exception with galangin (IC_50_ = 159 μM). It had a single hydroxyl group at C3 site which displayed moderate inhibition potential on thrombin compared to chrysin (without C3-OH, IC_50_ > 500 μM,). The absence of steric hindrance effects on B ring on galangin without any substituents made it easy for the flavonoid to enter the interior of the thrombin and bind the C3-OH to the protein.

Introduction of O-glycosyl groups at the C3 site on flavonoids weakened or led to loss of thrombin inhibitory activity. For instance, myricetin (IC_50_ = 56.6 μM) vs. myricitrin (IC_50_ = 79.5 μM); kaempferol (IC_50_ = 107 μM) vs. astragaline (IC_50_ = 217μM); quercetin (IC_50_ = 205 μM) vs. rutin (IC_50_ = 273 μM) and quercitrin (IC_50_ > 500 μM) and quercetin-3-*O*-β-d-glucoside (IC_50_ > 500 μM). Moreover, Isorhamnetin (IC_50_ = 72.2 μM) vs. isorhamnetin-3-*O*-neohespeidoside (IC_50_ > 500 μM) and typhaneoside (IC_50_ > 500 μM). Furthermore, flavonoid docking into the active center of 2GDE was hindered due to steric effects on the C3 site caused by the introduction of hydroxy group and glycosyl group at the C3 site. 

#### 2.5.2. The Influence of Hydroxyl, Methoxyl and O-glycosyl Groups at the C5, C6, C7, C8 Sites of A Ring

Thrombin inhibition was higher with trihydric phenol hydroxyl groups at the C5, C6 and C7 sites compared to meta-phenolic hydroxyl at the C5, C7 sites. In addition, more OH groups in the A-ring are beneficial for inhibitory effect [17]. However, the phenomenon was broken by the O-methylation at the C6 or C7 or C8 site. For example; scutellarein (C5, 6, 7-OH, IC_50_ = 70.8 μM) vs. apigenin (C5, 7-OH, without C6-OH, IC_50_ = 96.2 μM) vs. hispidulin (C5, 7-OH, C6-OCH_3_, IC_50_ = 125 μM) vs. genkwanin (C5-OH, C7-OCH_3_, IC_50_ = 212 μM). Also, baicalein (C5, 6, 7-OH, IC_50_ = 249 μM,) vs. chrysin (C5, 7-OH, IC_50_ > 500 μM) vs. wogonin (C5, 7-OH, C8-OCH_3_, IC_50_ > 500 μM). In another instance, luteolin (C5, 7-OH, IC_50_ = 146 μM) vs. 6-methoxyluteolin (C5, 7-OH, C6-OCH_3_, IC_50_ > 500 μM). There was an exception with hydroxygenkwanin (C5-OH, C7-OCH_3_, IC_50_ = 99.7 μM) with an OCH_3_ at the C7 site that had a higher activity than luteolin (C5, 7-OH, IC_50_ = 146 μM). Moreover, a hydroxyl at the C5 site weakened the anti-thrombin activity in fisetin (C7-OH, IC_50_ = 175 μM) vs. quercetin (C5, 7-OH, IC_50_ = 205 μM). This could be attributed to hydrogen-bond interaction between C5-OH and C4 = O on the quercetin. Moreover, an ortho-hydroxyl substitution on the C7 site near the C8-OH site had a moderate influence on inhibitory activity; for instance, kaempferol (C5, 7-OH, without C8-OH) vs. herbacetin (C5, 7, 8-OH) with the IC_50_ values 107 μM and 133 μM, respectively. 

The thrombin inhibitory activity of flavones and flavanones increased when OH group at the C7 site was replaced by O-glycosyl group. For example, baicalin vs. baicalein with IC_50_ values of 88.6 μM and >500 μM, respectively. In the case of flavonol, there was an exception. Kaempferol-7-*O*-β-d-glucopyranoside with an O-glycosyl groups at the C7 site was inactive to thrombin with IC_50_ > 500 μM compared with kaempferol (IC_50_ = 107 μM). Glycosylation at the 7 position on the ring system might have affected activity in a different way to glycosylation at C3. 

#### 2.5.3. The Influence of Hydroxyl and Methoxyl Groups at the C2′, C3′, C4′ and C5′ Sites of B Ring

Flavonoids with single hydroxy group at the C4′ site showed relative high activity of thrombin inhibition activity. IC_50_ values of Kaempferol (C4′-OH) vs. galangin (without C4′-OH) were 107 μM vs. 159 μM whereas scutellarein (C4′-OH) vs. baicalein (without C4′-OH) had IC_50_ values 70.8 μM vs. 249 μM. The 3D binding conformation of scutellarein (C4′-OH) indicated that C4′-OH interacted with residues ASPH189 (Figure 2B).

Compared to the single hydroxy group at the C4′ site, ortho-diphenol hydroxyl groups of ortho-position at the C3′, C4′ sites and meta-diphenol hydroxyl groups at the C2′, C4′ sites reduced thrombin inhibition activity. For instance, apigenin (C4′-OH, IC_50_ = 96.2 μM) vs. luteolin (C3′, 4′-OH, IC_50_ = 146 μM). In another example, kaempferol (C4′-OH, IC_50_ = 107 μM) vs. quercetin (C3′, 4′-OH, IC_50_ = 205 μM) vs. morin (C2′, 4′-OH, IC_50_ = 237 μM). However, there was an exception with hydroxygenkwanin (C3′, 4′-OH, IC_50_ = 99.7 μM). It had diphenol hydroxyl groups at the C3′, C4′ sites which exhibited higher activity compared with genkwanin (C4′-OH, IC_50_ = 211 μM). This exception may be due to the influence of C8-OCH_3_, compared with three hydroxyl substitutions (apigenin), the additional introduction of OCH_3_ (hydroxygenkwanin) increased the anti-thrombin activity. Triphenol hydroxyl groups at the C3′, C4′, C5’ sites were very beneficial for effecting thrombin inhibition activity. Myricetin (C3′, C4′, C5′-OH) vs. kaempferol (C4′-OH) vs. quercetin (C3′, 4′-OH) reported IC_50_ values, 56.5Μm, 107 μM and 205 μM, respectively. 

The primary SARs of natural flavones against thrombin are summarized in Figure 3. In conclusion, all these findings demonstrated that the C2=C3 and C4=O bonds of flavones were essential for thrombin inhibition. Thrombin inhibitory activity of flavonols and flavonoids was higher whereas that of flavanonols, flavanones and catechins was almost inactive. The introduction of OH groups of different quantities and at different position on flavonoids had diverse influence on thrombin inhibition. According to the bonding site (Figure 2), OH groups on C5, C7, C3′, C4′ and C5′ site on A ring and B ring were beneficial for anti-thrombin effect. In addition, introduction of O-methylation and O-glycosyl groups mainly played a role in weakening anti-thrombin activity.

## 3. Materials and Methods

### 3.1. Materials and Chemicals

Chromogenic substrate S-2238 was purchased from Chromogenix (Milan, Italy); bovine thrombin and argatroban were purchased from Sigma-Aldrich (St. Louis, MO, USA). The phosphate buffer saline (PBS) (pH 7.2-7.4, 0.01 M) and dimethyl sulfoxide (DMSO) were purchased from Solarbio (Beijing, China). Standard substance of galangin, kaempferol, astragaline, fisetin, quercetin, quercitrin, isoquercitrin, rutin, isorhamnetin, isorhamnetin-3-*O*-neohespeidoside, typhaneoside, myricetin, myricitrin, baicalein, baicalin, wogonin, wogonoside, hispidulin, scutellarein, genkwanin, hydroxygenkwanin, luteolin, luteoloside, 6-methoxyluteolin, dihydroquercetin, dihydromyricetin, naringenin, naringin, narirutin, hesperetin, hesperidin, neohesperidin, L-epicatechin, (-)-epigallocatechin, (-)-epicatechin gallate, proanthocyanidin B1 with 98% purity on the basis of HPLC analysis were obtained from Chengdu Herbpurify Co., Ltd. (Chengdu, China). Herbacetin and morin were obtained from Chengdu Must Bio-Technology Co., Ltd. (Chengdu, China). chrysin, apigenin, diosmetin and kaempferol-7-*O*-β-d-glucopyranoside were purchased from Chengdu Desite Bio-Technology Co., Ltd. (Chengdu, China).

### 3.2. In Silico Molecular Docking

#### 3.2.1. The Pretreatment of Receptor and Ligands

The crystal structure of thrombin (2GDE, PBD ID) and target protein were downloaded from the PBD database and pre-processed by Discovery Studio (DS) V 4.5 (Accelrys, Biovia, Boston, MA, USA) including addition of hydrogen atoms, protonation treatment, and deletion of water molecules, original ligand molecule (SN3) and other unrelated protein conformations. Then the crystal structure was energy optimized by adding CHARMm force field. The active pockets (sphere) were defined and radius modified to 9.0 Å based on its original ligand position.

All the collected compounds (103 flavonoids) from traditional Chinese herb medicines were sketched by ChemDraw V10.0 (Cambridge Soft, Cambridge, MA, USA). These compound structures were imported into DS, and then, CHARMm force field was added to minimize the energy.

#### 3.2.2. Molecular Docking and Virtual Screening

According to the above conditions, the compound structures and 2GDE were imported into DS for screening the potential flavonoids components which could act on thrombin. Docking results of 2GDE and ligand molecule were graded by “-CDOCKER_energy” and “-CDOCKER_INTERACTION energy”.

### 3.3. Thrombin Inhibitory Activity Assay

Activity of thrombin was measured on Multiskan™ FC microplate photometer (Thermo Scientific, Waltham, MA, USA) based on published method with modification to suit our requirements [4,19,23]. Briefly, 80 μL thrombin solution (14 μg/mL (46 U/mg) in phosphate buffer saline, pH 7.2-7.4, 0.01 M) was pre-incubated in 96-well plates with 10 μL of each flavonoid solution (1–10 mM in DMSO) and 90 μL PBS at 37 °C for 5 min and shaken for 1 min. Twenty microliter (20 μL) chromogenic substrate S-2238 (4 mg/mL in PBS) was then added to activate the reaction and incubated for 15 min at 37 °C after shaking for 1 min. All assays were paralleled three points and then measured at 405 nm in the micro-plate reader immediately. The blank group was 10 μL DMSO without flavonoids whereas Argatroban was used positive control. Background group was 10 μL flavonoid solutions (0.5–10 mM in DMSO) with 190 μL phosphate buffer solution. According to the pretest, 5% of DMSO in the total volume had no effect on thrombin. The thrombin inhibition rate was calculated using the following equation: Inhibition ratio (%) = [A_blank_ − (A_sample_ − A_background_)]/A_blank_× 100(1)
where A_blank_ was the absorbance of the blank group; A_sample_ was the absorbance of the sample group, and A_background_ was the absorbance of background group. IC_50_ values for active compounds were determined on three replicates of six concentrations. Statistical analyses and IC50 values were calculated using GraphPad Prism version 5.01 (GraphPad Software Inc., La Jolla, CA, USA).

## 4. Conclusions

In this study, potential active flavonoids were screened from traditional Chinese medicine by molecular docking; the activity was studied by enzymatic bioassay, and SARs of natural flavonoids against thrombin were revealed. Molecular docking provided a method of screening target compounds with specific structure from a large molecular dataset quickly and easily. The enzymatic bioassay focused on functional verification of thrombin inhibitors from the perspective of molecular biology. The combination of the two methods provided a reference to accurately explore the structural and activity characteristics of flavonoids against thrombin. In this paper, the results showed that the C2, C3 and C4 sites on C ring and C5, C7 sites on A ring as well as C3′, C4′ and C5′ sites on B ring were the key sites of flavonoids for activity against thrombin. Moreover, C=C, C=O bonds and OH groups were the indispensable substituent group for flavonoids against thrombin. In conclusion, these are the key aspects revealed through the SAR analysis of natural flavonoids on anti-thrombin activity. These findings would promote the study on activity mechanism of flavonoid-type compounds as well as aid medicinal chemists to design and develop more potent flavonoid-type inhibitors against thrombin.

## Figures and Tables

**Figure 1 molecules-25-00422-f001:**
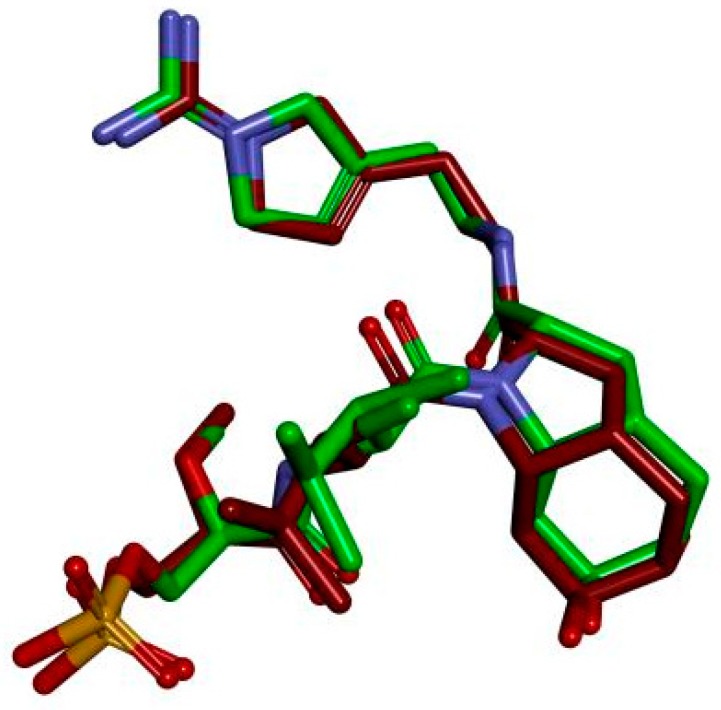
Conformation of re-docked co-crystallized thrombin complex (2GDE and SN3).

**Figure 2 molecules-25-00422-f002:**
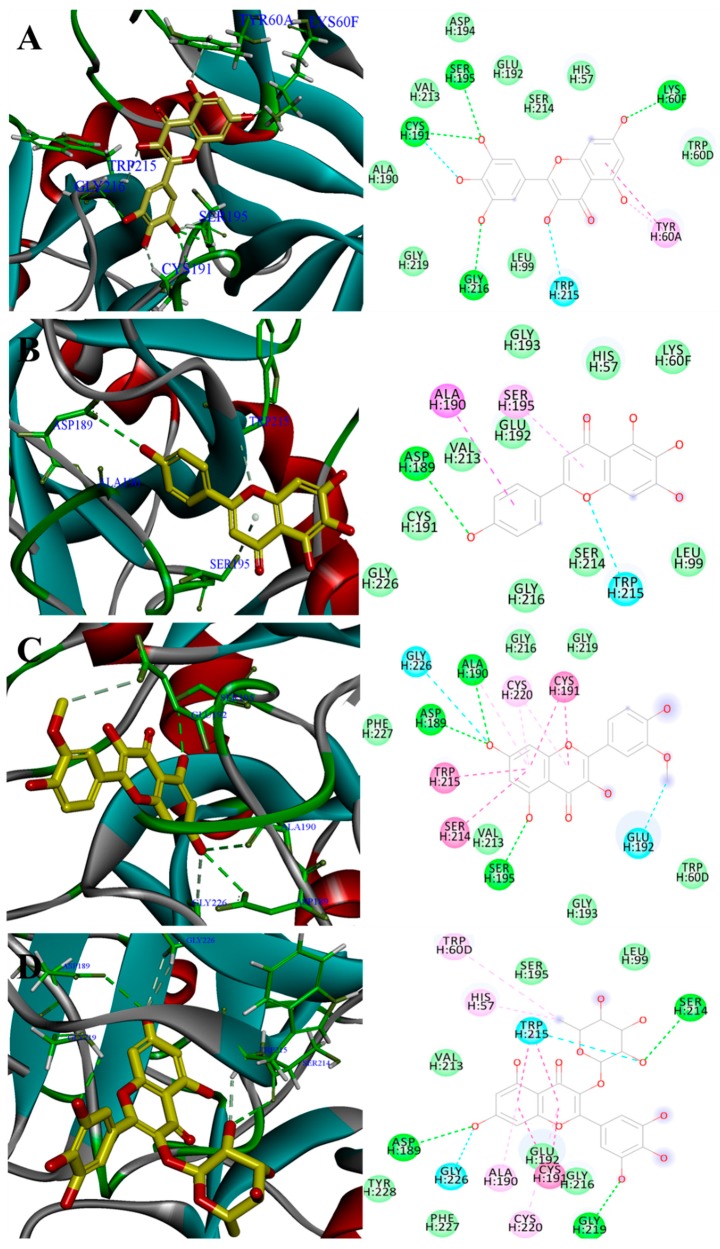
Binding conformation of 4 target flavonoids at the active site of thrombin: (**A**) Myricetin, (**B**) Scutellarein, (**C**) Isorhamnetin, (**D**) Myricitrin.

**Figure 3 molecules-25-00422-f003:**
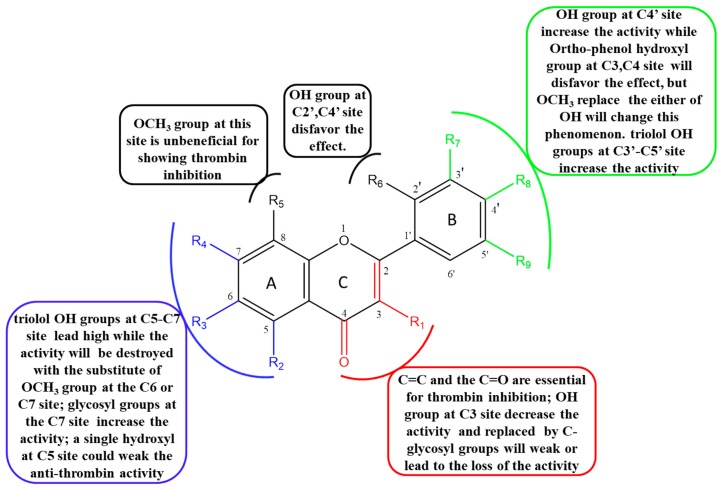
The structure-inhibition relationships of flavones against thrombin.

**Table 1 molecules-25-00422-t001:**
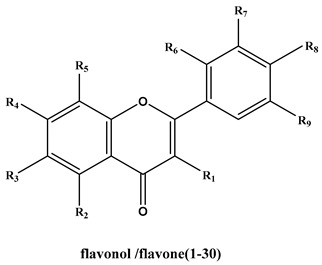
The structure, IC_50_ for anti-thrombin and docking score of 42 compounds.

No.	Compound	R1	R2	R3	R4	R5	R6	R7	R8	R9	IC_50_(μM)	-CDOCKER_Energy(Kcal/mol)	-CDOCKER_INTERACTION_Energy(Kcal/mol)
**flavonol**
**1**	Galangin	OH	OH	H	OH	H	H	H	H	H	159 ± 0	27.5	32.1
**2**	Herbacetin	OH	OH	H	OH	OH	H	H	OH	H	133 ± 4	31.0	31.2
**3**	Kaempferol	OH	OH	H	OH	H	H	H	OH	H	107 ± 2	33.6	40.2
**4**	Astragaline	O-glc	OH	H	OH	H	H	H	OH	H	217 ± 5	7.92	46.8
**5**	Kaempferol-7-*O*-β-D-glucopyranoside	OH	OH	H	O-glc	H	H	H	OH	H	>500	22.6	48.5
**6**	Morin	OH	OH	H	OH	H	OH	H	OH	H	237 ± 26	33.0	39.3
**7**	Fisetin	OH	H	H	OH	H	H	OH	OH	H	175 ± 18	33.4	36.2
**8**	Quercetin	OH	OH	H	OH	H	H	OH	OH	H	205 ± 14	32.2	34.3
**9**	Quercitrin	O-rha	OH	H	OH	H	H	OH	OH	H	>500	17.5	51.2
**10**	Isoquercitrin	O-glc	OH	H	OH	H	H	OH	OH	H	>500	13.6	58.8
**11**	Rutin	O-rut	OH	H	OH	H	H	OH	OH	H	274 ± 23	5.76	61.7
**12**	Isorhamnetin	OH	OH	H	OH	H	H	OCH_3_	OH	H	72.2 ± 4.8	29.2	35.1
**13**	Isorhamnetin-3-*O*-neohesperidoside	O-glc-rha	OH	H	OH	H	H	OCH_3_	OH	H	>500	5.76	57.2
**14**	Typhaneoside	O-rha-(glc)2	OH	H	OH	H	H	OCH_3_	OH	H	>500	2.97	75.8
**15**	Myricetin	OH	OH	H	OH	H	H	OH	OH	OH	56.5 ± 2.1	36.2	37.7
**16**	Myricitrin	O-rha	OH	H	OH	H	H	OH	OH	OH	79.5 ± 3.4	17.4	50.3
**flavone**
**17**	Chrysin	H	OH	H	OH	H	H	H	H	H	>500	25.5	30.3
**18**	Baicalein	H	OH	OH	OH	H	H	H	H	H	249 ± 31	28.7	28.5
**19**	Baicalin	H	OH	OH	O-glc	H	H	H	H	H	88.6 ± 8.2	20.5	40.0
**20**	Wogonin	H	OH	H	OH	OCH_3_	H	H	H	H	>500	20.1	28.6
**21**	Wogonoside	H	OH	H	O-glu	OCH_3_	H	H	H	H	>500	29.5	38.2
**22**	Hispidulin	H	OH	OCH_3_	OH	H	H	H	OH	H	126 ± 6	29.5	38.2
**23**	Scutellarein	H	OH	OH	OH	H	H	H	OH	H	70.8 ± 2.7	35.1	35.7
**24**	Apigenin	H	OH	H	OH	H	H	H	OH	H	96.2 ± 10.0	25.9	30.8
**25**	Genkwanin	H	OH	H	OCH_3_	H	H	H	OH	H	212 ± 30	26.1	33.2
**26**	Hydroxygenkwanin	H	OH	H	OCH_3_	H	H	OH	OH	H	99.7 ± 5.5	33.6	38.6
**27**	Luteolin	H	OH	H	OH	H	H	OH	OH	H	146 ± 10	31.5	36.6
**28**	Luteoloside	H	OH	H	O-glc	H	H	OH	OH	H	155 ± 18	14.1	45.4
**29**	Diosmetin	H	OH	H	OH	H	H	OH	OCH_3_	H	131 ± 14	31.41	38.4
**30**	6-Methoxyluteolin	H	OH	OCH_3_	OH	H	H	OH	OH	H	>500	31.0	38.7
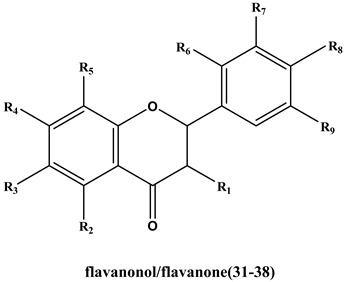
**flavanonol**
**31**	Dihydroquercetin	OH	OH	H	OH	H	H	OH	OH	H	>500	34.4	39.7
**32**	Dihydromyricetinflavanone	OH	OH	H	OH	H	H	OH	OH	OH	>500	40.3	43.3
**33**	Naringenin	H	OH	H	OH	H	H	H	OH	H	>500	30.3	35.5
**34**	Naringin	H	OH	H	O-glu-rha	H	H	H	OH	H	444 ± 59	7.6	56.3
**35**	Narirutin	H	OH	H	O-rut	H	H	H	OH	H	>500	10.8	55.7
**36**	Hesperetin	H	OH	H	OH	H	H	OH	OCH_3_	H	>500	35.5	41.9
**37**	Hesperidin	H	OH	H	O-glu-rha	H	H	OH	OCH_3_	H	>500	6.28	60.1
**38**	Neohesperidin	H	OH	H	O-(glc)2	H	H	OH	H	OCH_3_	>500	8.64	55.2
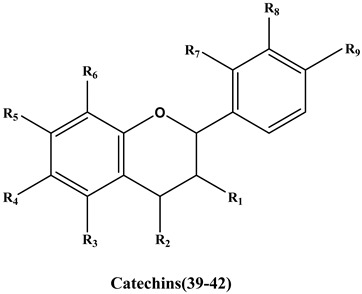
**Catechins**
**39**	Epicatechin	OH↑	H	OH	H	OH	H	H	OH	OH	>500	31.5	36.5
**40**	(-)-epigallocatechin	OH↓	H	OH	H	OH	H	H	OH	OH	>500	33.4	39.3
**41**	(-)-Epicatechin gallate	1*	H	OH	H	OH	H	H	OH	OH	>500	52.1	54.6
**42**	Proanthocyanidin B1	OH	2*	OH	H	OH	H	H	OH	OH	>500.0	42.4	55.9
	Argatroban	-	-	-	-	-	-	-	-	-	1.86 ± 0.10	-	-

1* 
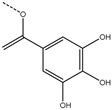
; 2* 
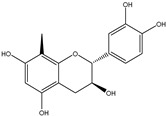
; glc, glucose; glu, glucuronic acid; rha, rhamnose; rut, rutinose.

**Table 2 molecules-25-00422-t002:** Theoretical binding site in thrombin model.

Group	Compound	IC_50_ ±SD (μM)	-Cdocker_ Energy	-Cdocker_ Interaction_ Energy	Number of Hydrogen Bonds	Residue
Strong inhibitor	Myricetin	56.5 ± 2.1	36.2	37.7	5	LYS60F, TYR60A, SER195, CYS191, GLY216
	Scutellarein	70.8 ± 2.7	35.1	35.7	4	ALA190, ASP189, TRP215, SER195
	Isorhamnetin	72.2 ± 4.8	29.2	35.1	9	ALA190, ASP189, GLY226, GLU192, SER195, TRP215, SER214, CYS191, CYS220
	Myricitrin	79.5 ± 3.4	17.4	50.3	10	GLY226, ASP189, SER214 *, GLY219, TRP215 *, ALA190, CYS191, CYS191, HIS57 *, TRP60D *
	Baicalin	88.6 ± 8.2	20.5	40.0	5	ALA190, CYS191, TRP215, GLY193, SER195
	Apigenin	96.2 ± 10.0	25.9	30.8	2	TRP60D, TYR60A
	Hydroxygenkwanin	99.7 ± 5.5	33.6	38.6	3	HIS54, LYS60F, SER195
Moderate inhibitor	Kaempferol	107 ± 2	33.6	40.2	6	CYS220, CYS191, SER214, TRP215, ALA190, GLY226
	Hispidulin	126 ± 6	29.5	38.2	7	TRP60D, SER214, TRP215, CYS191, ALA190, HIS57, SER195
	Diosmetin	131 ± 14	31.41	38.4	5	SER195, TYR60A, HIS57, LYS60F, HIS57,
	herbacetin	133 ± 4	31.0	31.2	5	CYS191, TRP215, GLU192, HIS57, TYR60A,
	Luteolin	146 ± 10	31.5	36.6	6	CYS220, CYS191, SER214, TRP215, ALA190, GLU192
	luteoloside	155 ± 18	14.1	45.4	-	-
	galangin	159 ± 0	27.5	32.1	5	HIS57, HIS57, ALA190, CYS220, CYS191
	fisetin	175 ± 18	33.4	36.2		CYS191, SER195, LYS60F, TYR60A, TRP215

*: the H-bonding interactions of 2GDE and the OH groups of sugars for myricitrin.

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
