# Peer review of "Study on Structure Activity Relationship of Natural Flavonoids against Thrombin by Molecular Docking Virtual Screening Combined with Activity Evaluation In Vitro"

_molecules, 2020, doi:10.3390/molecules25020422_

Round 1

Reviewer 1 Report

My comments are indicated on teh PDF copy.

Author Response

We thank the reviewer for these insightful suggestions and our responses to the comments are listed as follow:

Page 1 – Line 32 – 50 should be subscript throughout.

Respond: “50” has been subscript throughout (Line 32).

Page 2 – Line 74 –“provided” should be corrected as “provide”. Page 2 – Line 73 –This sentence would fit well the last part of Conclusions.

Respond: “provided” has been corrected as “provide” (Line 79). Consider that the last sentence in the conclusion has the same meaning as this sentence, so this sentence has been corrected as “The findings of this research aim to provide a basis…” (Line 78-79).

Page 8 - Line 144 –“were” should be corrected as “are”.

Respond: “were” has been corrected as “are” (Line 143).

Page 10 – Line 165 –“inhibition thrombin” should be corrected as “on inhibition of thrombin”.

Respond: “inhibition thrombin” has been corrected as “on inhibition of thrombin” (Line 161).

Page 11 – Line 208 –“activity. For” should let this be a continuous sentence.

Respond: “activity. For” has been merged into a continuous sentence as “activity, for” (Line 205).

Page 11 – Line 232,263 –“recorded” should be corrected as “exhibited”.

Respond: “recorded” has been corrected as “exhibited” (Line 225).

Page 11 – Line 239 –revisit this statement. It is confusing and loses the meaning.

Respond: I’m sorry that this statement is an edit error, and the sentence has been rewritten as “This exception may be due to the influence of C8-OCH3, compared with three hydroxyl substitutions (apigenin), the additional introduction of OCH3 (hydroxygenkwanin) increased the anti-thrombin activity” (Line 226-228).

Page 11 – Line 240 –is 'effecting or causing' not appropriate.

Respond: “showing” has been corrected as “effecting” (Line 228).

Page 13 – Line 298 –The highlighted ([Ablank - (Asample- Abackground)]/ Ablank× 100) should be subscripts.

Respond: The highlighted has been subscripts (Line 283).

Page 13 – Line 304 –Though not incorrect, the statement would sound reader-friendly if rephrased to start as follows: SAR of natural flavonoids... through the combination of molecular docking and enzymatic bioassay.

Respond: The sentence has been corrected as “In this study, potential active flavonoids were screened from traditional Chinese medicine by molecular docking, the activity was studied by enzymatic bioassay, and SAR of natural flavonoids against thrombin was revealed” (Line 289-291).

Page 13 – Line 310 –“and” should be corrected as “as well as”.

Respond: “and” has been corrected as “as well as” (Line 295).

Page 13 – Line 310 –for activity against thrombin. Otherwise the sentence sounds incomplete.

Respond: “for activity against thrombin” has been add in the end of this sentences (Line 296).

Page 13 – Line 312 –“In conclusion, these key findings discussed the SARs of natural flavonoids anti-thrombin activity” should be corrected as “In conclusion, these are the key aspects revealed through the SAR analysis of natural flavonoids on anti-thrombin activity”.

Respond: This sentence has been corrected as “In conclusion, these are the key aspects revealed through the SAR analysis of natural flavonoids on anti-thrombin activity” (Line 297-299).

Page 14 – Line 347 –references should be start with surname for consistency.

Respond: This reference (12) has been corrected as started with surname (Line 333).

Page 14 – Line 361 –“Pawel” seems like first name and should be abbreviated and follow the surname.

Respond: “Pawel” of reference (19) has been abbreviated and follow the surname (Line 348).

Page 14 – Line 365 –references should be start with surname for consistency.

Respond: This reference (21) has been corrected as started with surname (Line 352).

Reviewer 2 Report

This manuscript developed a method by combining molecular docking and chromogenic substrate method to reveal the Structure Activity Relationship (SAR) of natural flavonoids against thrombin. In fact, a number of published articles have used molecular docking to study the interaction between small molecule compounds and thrombin. And there was also a literature report on the SAR of flavonoids inhibiting thrombin. But this article carried out the molecular docking-based screening of 103 flavonoids for the first time and investigate anti-thrombin activity for all the substituent group to reveal the SARs of natural flavonoids against thrombin. In summary, this work was significant but not novel enough. Here give some suggestions for further revision:

Chromogenic substrate method should be described in detail in the introduction. The "virtual screening" in this study only includes the part of molecular docking, is it too simple? Can the author consider adding conditions, such as the Lipinski 5 principle, and performing ADME prediction calculations? Pay attention to some mistakes, for example, “IC50” should be written out as “IC50”, “37oC “ should be changed to “37°C “(line 291), “–cdocker_ and –cdocker_interaction energies” may be changed to “-CDOCKER_energy and -CDOCKER_ INTERACTION energy”(line 166)? Please check the whole manuscript carefully. “thrombin activity” should be changed to “anti-thrombin activity”. (line 63) Please adjust the sharpness of all pictures in the

Author Response

We thank the reviewer for these insightful suggestions and our responses to the comments are listed as follow:

Chromogenic substrate method should be described in detail in the introduction.

Respond: The Chromogenic substrate method has been described in detail in the introduction, “This assay determines the activity of enzymes in samples by measuring the absorbance of coloured p-nitroaniline produced by the reaction of blood coagulation factor (such as FX, thrombin and prothrombin) and chromogenic substrate [24]. Clinically, CSA is widely used in diagnosing and monitoring of hemophilia A/B, investigating anticoagulant effects and screening enzyme inhibitors [5, 25, 26]” (Line 71-75).

The "virtual screening" in this study only includes the part of molecular docking, is it too simple? Can the author consider adding conditions, such as the Lipinski 5 principle, and performing ADME prediction calculations?

Respond: Thank the reviewer for the insightful comment. In our study, the potential active flavonoids were screened from Chinese herb medicine. Due to the particularity of Chinese herb medicine, some active compounds would be missed if perform ADME prediction (such as Lipinski 5 principle). For example, ginsenosides are the main medicinal components of ginseng. However, the molecular weight of many ginsenosides is greater than 500, and the number of hydrogen bond donors is greater than 5. If Lipinski 5 principle was performed before molecular docking, some active components would be missed. Our experiments are basic research, and the purpose of which is to screen out the potential active flavonoids and study the structure activity relationship, not considering the druggability. We will consider ADME prediction calculations and modify the structure of the compound at a later stage, if it has a better activity.

Pay attention to some mistakes, for example, “IC50” should be written out as “IC50”, “37oC “ should be changed to “37°C “(line 291), “–cdocker_ and –cdocker_interaction energies” may be changed to “-CDOCKER_energy and -CDOCKER_ INTERACTION energy”(line 166)

Respond: “IC50” in whole manuscript has been corrected to “IC50”, “37oC” has been corrected to “37°C” (line 276), “–cdocker_ and –cdocker_interaction energies” has been changed to “-CDOCKER_energy and -CDOCKER_ INTERACTION energy” (Line 162).

Please check the whole manuscript carefully. “thrombin activity” should be changed to “anti-thrombin activity”. (line 63)

Respond: “thrombin activity” had be changed to “anti-thrombin activity” (Line 63).

Please adjust the sharpness of all pictures in the

Respond: The sharpness of all pictures in the manuscript had been adjusted.

Reviewer 3 Report

This manuscript requires significant revision:

see attached comments

Author Response

We thank the reviewer for these insightful suggestions and our responses to the comments are listed as follow:

The manuscript requires significant revision of English expression (particularly regarding inclusions of definite and indefinite articles tenses and singular/plural issues)

Respond: The English expression (particularly regarding inclusions of definite and indefinite articles tenses and singular/plural issues) had been revised carefully.

Careful analysis of the estimated precision of the IC50 values presented in Table 1.

Respond: The estimated precision of the IC50 values presented in Table 1 has been corrected.

IC50 values and CH3 groups should have the numbers subscripted throughout the entire manuscript.

Respond: The numbers of IC50 values and CH3 groups has been subscripted throughout the entire manuscript.

Specific responses to the comments are listed below:

Page 1 – Line 23 –“…in the blood coagulation cascade…”

Respond: “the” has been added (Line 22).

– Line 24–:  “…relationships.”

Respond: “relationship” has been corrected as “relationships” (Line 25).

– Line 32–:  “…IC50 of 56 M …”

Respond: “…IC50 of 56.45 μM …” has been corrected as “…IC50 of 56 μM …”(Line 32).

–Line 33-34–: “..is not limited to exploring flavonoid structure activity relationships to anti-thrombin activity….”

Respond: “..is not limited to only exploring the structure activity relationship flavonoid’s anti-thrombin activity…” has been corrected as “.. exploring flavonoid structure activity relationships to….”(Line 34).

5.Page 2 – Line 46 –“…discover the safer…”

Respond: “the” has been deleted (Line 46).

– Line 49–: cyaniding-3-glugoside

Respond: The “g” of cyaniding-3-glugoside has been deleted (Line 49).

– Line 56–: “…but are not…”

Respond: “are” has been add (Line 56)

– Line 69–: “A chromogenic substrate…”

A reference to publication of this methodology should be included here: you have in the experimental section (Page 13, line 289) - … "based on the modified method [4, 19, 24]." Also this means references 23 and 24 will need to be switched in your reference list. To what does "modified" refer? Do your references use a method modified from a previous publication, or have you modified the published method to suit your requirements?

Respond: “A chromogenic substrate…”,  “A” has been add (Line 69).

The related reference to publication of this methodology has been including (Line 71). The order of the reference has been rearranged (23, 27) (Line 357, 367). The published method has been modified to suit our requirements, so this sentence has been corrected as “…based on the published method with modification to suit our requirements” (Line 273).

– Line 71: “…of the chromogenic…”

 Respond: “the” has been adding (Line 75).

6.Page 3 – Line 92 –“..the activities of 42 target flavonoids were selected for further….”

Respond: “activity” has been corrected as “activities” , “was” has been corrected as “were”,  “for” has been adding (Line 97).

– Line 93–: “…using a chromogenic substrate in vitro method.

Respond: this sentence has been corrected as “…using a chromogenic substrate in vitro method” (Line 98).

– Line 98–: “…chemical constructions structures of…”

Respond: “constructions” has been corrected as “structures” (Line 103)

7.Page 4 – Line 114 –and column heading: IC50

Respond: The numbers of IC50 values has been subscripted (Line 120).

– Line 102–: “…flavonoids are listed…”

Respond: The “were” has been corrected as “are” (Line 107).

8.Page 6 and 7 column heading: IC50

Respond: The numbers of IC50 values has been subscripted (Line 123).

9.Page 8 – Line 125-6,132,134-5 – IC50

Respond: The numbers of IC50 values has been subscripted (Line 126-131).

– Line 135–: “…binding sites…

Respond: The “site” has been corrected as “sites” (Line 139).

– Line 136–: “…is shown…”

Respond: “was” has been corrected as “is” (Line 140)

– Line 136-141–: As I see it, all the OH group H-bonding interactions for the 4 flavonoid nuclei are indicated in either or both Table 2 and Figure 2, so the information presented here should be omitted, as it simply replicates information presented elsewhere. However, it would be useful to differentiate the H-bonding interactions of the OH groups on the flavonoid nucleus in the binding site, from those of the sugars for isorhamnetin and myricitrin presented here, perhaps by asterisks for the sugar interactions and the addition of an explanatory footnote to both Table 2 and Figure 2.

I presumed the interactions between ring A of myricetin and Tyr H:60A and  those between rings A/B of isorhemnetin and myricitrin and Try H:215  were probably a -interactions, but what are the interactions shown for the B and C rings of scutellarin with Ser H:195 and Ala H:190 in Figure 2…and why haven’t they been mentioned.

Respond: Thank the reviewer for pointing this replicates information and the repetitive information presented here has been omitted. I’m sorry that the Try H:215, Ser H:195 and Ala H:190 has been omitted duo to an edit error. In the both Table 2 and Figure 2, it clear that the Tyr H:60A interacted with myricetin, Try H:215 interacted with isorhemnetin and myricitrin, as well as Ser H:195 and Ala H:190 interacted with scutellarin.

– Line 144–: “ The binding conformations ……are shown…”

The section between lines 144 and line 158 also requires careful revision to remove duplication of information, but meaningful structure-activity discussion material should be retained.

Respond: “conformation” has been corrected as “conformations” and “are” has been adding (Line 143). This repetitive information has been removed and this sentence has revision (see the red section in Line 144-157).

Page 10 – Line 174,178 – IC50 = 56.5 μM

Respond: The numbers of IC50 values has been subscripted (line 170-175). “IC50 = 56.45 μM” has been corrected as “IC50 = 56.5 μM” (line 171).

– Line 179–: 96.3 μM

Respond: “IC50 = 96.26 μM” has been corrected as “IC50 = 96.3 μM” (Line 176).

– Line 198–: “…substitute of O-methylation at…”; also “scutellarein”

Respond: “substitute of” has been deleted, “sutellarein” has been corrected as “scutellarein” (Line 196)

– Line 202–: “6-methoxyluteolinv”

Respond: “6-methoxyluteolinv” has been corrected as “6-methoxyluteolin” (Line 199)

Page 11 – Line 208 –“…had a moderate influence…”

Respond: “a” has been adding (Line 205).

– Line 211-214–: The thrombin inhibitory activity of flavones and flavanones increased when OH group at the C7 site was replaced by O-glycosyl group. For example, baicalin vs baicalein with IC50 values of 88.55 μM and >500 μM, respectively and naringin (IC50 value = 444 μM) vs naringenin (IC50 value = 249 μM).

The example (highlighted in red) as presented contradicts the conclusion it is supposed to support: Naringin (the glycoside) (IC50 = 444 μM) is LESS active than Naringenin (the aglycone) (IC50 = 249 μM), not MORE active. BUT, that is because the IC50 for naringenin isn’t 249 μM: it is >500 μM in Table 2! It’s not really a very good example to use though, as the SD for naringin of ± 59 M means its IC50 might also be >500 M, so statistically not significantly different from the data presented for naringenin.

Respond: The IC50 =249 μM for naringenin is an edit error. I strongly agree with reviewer for the insightful comment that statistically not significantly different from the data presented for naringenin and naringin. So this example has been deleted.

– Line 216-217–: “This inactivity could be due to the different of C3 site hydroxyl substitution.” (?) What the authors are trying to say here isn’t at all clear to me, but I can guess they might be trying to say that glycosylation at the 7 position on the ring system might affect activity in a different way to glycosylation at C3.

Respond: Thank the reviewer for pointing this. This original sentence is not clear and is has corrected as “Glycosylation at the 7 position on the ring system might have affected activity in a different way to glycosylation at C3” (Line 211-212).

– Line 220-221–: “…were 107μM vs 159μM”

Respond: “158.6” has been corrected as “159” (Line 216).

– Line 224–: “Compared to the single hydroxyl group at the C4’ site, ortho-diphenol

hydroxyl groups at the C3’, C4’ sites and meta-diphenol hydroxyl groups at the C2’, C4’ sites …”

Respond: “Comparing with” has been corrected as “Compared to”, “diphenol hydroxyl” has been corrected as “ortho-diphenol hydroxyl”, “meta-position” has been corrected as “meta-diphenol hydroxyl groups” (Line 219-220).

– Line 226-227–: “However, replacement of either of the OH group at the C3’ site or C4’ site in the B-ring by an OCH3 group increased thrombin inhibition.

Respond: Duo the example of diosmetin and luteolin is no suitable, so this sentence has been deleted.

– Line 228–: “…luteolin (C3’, 4’-OH, IC50=146 μM) vs diosmetin (C3’-OH, C4’-OCH3, IC50=131 μM)”.

Given that the SD values on these two IC50 values are ± 10μM and ±15μM respectively and the difference between them is only 15μM, these values are not statistically significantly different.

Respond: I strongly agree with and thank the reviewer for the insightful comment. There are not statistically significantly different for only 15μM differences between the two compounds and it doesn't make sense to put the example in this place. So this example of diosmetin and the collusion of “replacement of either of the OH group at the C3’ site or C4’ site in the B-ring replaced by an OCH3 group increased thrombin inhibition” have been deleted.

– Line 232–: Why has a structure number (25) appeared here for the first time in the manuscript before the bracket end?

Respond: “25” is an edit error and it has been deleted.

– Line 233-239–: This repeats the same “sentence” twice, and I have no idea what it is about: it needs to be rewritten in English.

Respond: I’m sorry that the repeats the same “sentence” are an edit error, and the sentence has been rewritten as “This exception may be due to the influence of C8-OCH3, compared with three hydroxyl substitutions (apigenin), the additional introduction of OCH3 (hydroxygenkwanin) increased the anti-thrombin activity” (line 225-228).

– Line 241–: Two more structure numbers?? Why? There is no consistency in the style here.`

The remainder of page 11 needs to be carefully examined/ revised, as I am sure the effects of 3’,4’,5’-trihydroxyflavones including myricetin and its binding were previously presented. This discussion has become more disorganised the further the reader progresses into it.

Respond: The structure numbers is an edit error and it has been deleted. The conclusion of 3’, 4’, 5’-trihydroxyflavones was discussed from the influence of C4’ site, C3’, C4’ sites and 3’, 4’, 5’site, respectively, so 3’,4’,5’-trihydroxyflavones is not repeated here. The binding of myricetin has been deleted (Line 228-230).

12.Page 12 – Line 268 –“Herbacetin, morin were….”

Respond: “was” has been corrected as “were” (Line 253).

– Line 270–: Why does “Kaempferol-7-O-β-D-glucopyranoside” start with a capital letter when no other substance names are capitalised?

Respond: “Kaempferol-7-O-β-D-glucopyranoside” is an edit error and it has been lowercase (Line 254).

– Line 289–: see the previous comment about “modified method” related to line 69.

Respond: The published method has been modified to suit our requirements, so this sentence has been corrected as “…based on published method with modification to suit our requirements” (Line 273).

– Line 297–: “…calculated using the following equation:”

Respond: “inhibition rate was calculated as follow equation” has corrected as “calculated using the following equation” (Line 282).

13.Page 14 – Line 370-374 –These two references (23 and 24) need to be reversed to

compensate for the move of the former reference 24 to an earlier position in the discussion.

Respond: The order of the reference has been rearranged (Line357, 368).

Round 2

Reviewer 2 Report

The manuscript has been carefully revised and the comments have been well addressed. There is only one minor issue still needs to be confirmed. Please add theoretical binding sites for all the “strong inhibitors” and “moderate inhibitors”, at least in the Supplementary Materials.

Author Response

The manuscript has been carefully revised and the comments have been well addressed. There is only one minor issue still needs to be confirmed.

Please add theoretical binding sites for all the “strong inhibitors” and “moderate inhibitors”, at least in the Supplementary Materials.

Respond: We thank the reviewer for these insightful suggestions. The theoretical binding sites for all the “strong inhibitors” and “moderate inhibitors” have been added in the Table 2.

Reviewer 3 Report

Please see the attached report

Author Response

We thank the reviewer for these insightful suggestions and our responses to the comments are listed as follow:

These include incomplete and incorrect references; all ofwhich should now be thoroughly checked for completeness and accuracy

Respond: These references have been thoroughly checked and careful corrected followed the reviewer’s suggestions.

The supplementary material provided also needs to have the manuscript title, authors with affiliations, and a table of contents added

Respond: The manuscript title, authors with affiliations, and a table of contents have been added in the supplementary material.

Page 1 – Line 24 –“…inhibitors), and as well…”

Respond: The “and” has been deleted (Line 24).

Page 2 – Line 50 –“…eupatilin-7-O-β-D-glucopyranoside…”

Respond: The “d” has been corrected as “D” (Line 50).

– Line 70–: Italicise “in vitro

Respond: The “in vitro” has been italicized (Line 70).

5.Page 3 – Line 93 –“..showed that 62 hit compounds….”

Respond: The “that” has been deleted(Line 93).

– Line 106–: “IC50 values…”

Respond: The numbers of IC50 values has been subscripted (Line 106).

– Line 109-110–: (If they aren’t issues my computer has created) fix the formatting of

“Argatroban” to remove letter overlap, and delete space from “56.5”

Respond: There are no wrong with “Argatroban” in my computer, and the space from “56.5” has been deleted (Line 110).

– Line 113–: “Kaempferol”

Respond: The “f” has been adding (Line 113).

Page 4 – Line 119 – (Table caption): “Table 1. The structure, IC50 for anti-thrombin activity and docking scores of 42 compounds.”

Respond: “Table 1 the” has been corrected as “Table 1. The”. The numbers of IC50 values has been subscripted. “against thrombin” has been corrected as “anti-thrombin” (Line 119).

– Line 120–: Compound 3 in Table 1: “Kaempferol”

Respond: The “f” has been adding (Line 120).

7.Page 8 – Line 140–“…(2GDE) are shown…”

Respond: The “is” has been corrected as “are” (Line 138).

– Line 156–: “…hindered the some interactions between other sites and the target.”

Respond: This sentences “this hindered the bond between other sites and the target” has been corrected as “…hindered some interactions between other sites and the target” (Line 156).

Page 10 – Line 176,185 –“Kaempferol”

Respond: The “f” has been adding (line 176,184).

– Line 175-182–: As the apigenin SD is ±10 and the kaempferol SD is ±2, the

sum of SDs here (±12) exceeds the difference of the two values: 10.8, so this is not astatistically significant difference (but it is indicative). I think you can simply add a

few words to reduce the impact, but still tell the same story - I suggest:

"... flavonoids with single hydroxyl group at C3 site were observed to have slightly

decreased inhibitory activity. For example, apigenin (without C3-OH, IC50=96.3 μM)

vs kaempferol (with C3-OH, IC50=107 μM). Luteolin (without C3-OH, IC50=146 μM) vsquercetin (with C3-OH, IC50=205 μM)

Respond:  thanks for the insightful suggestion. This sentences has been corrected followed the reviewer’s suggestion (Line 174-175).

– Line 179–: “…displayed relatively strong moderate inhibition potential…”

Respond: The “relatively strong” has been corrected as “moderate” (Line 178)

Page 11 – Line 205,211,215,222 and 229 – and page 12– Line 247–“Kaempferol”

Respond: The “f” has been adding (Line 204,210,214,222,221).

10.Page 13 – Line 290-291 –“…and SARs of natural flavonoids against thrombin were revealed.”

Respond: The “s” has been adding and “was” has been corrected as “were” (Line289, 290).

11.Page 14 – Line 323 –The year, volume and pages for reference 7 should be 2018, 32,632-639.

Respond: The year, volume and pages have been corrected as “2018, 32,632-639” (Line 322).

– Line 335–: “…orientin in vitro, and in vivo.”

Respond: The “in vitro, and in vivo.” has been italic (Line 334).

– Line 340–: “J Appl Microbiol. 2019, 127, 1282-1290.”

Respond: The “127, 1282-1290.” Has been adding (Line 339).

– Line 344–: Thromb Res. 2010, 126, e365-e378.

Respond: The “e365-e378” has been adding (Line 343).

– Line 359–: “…in vitro and in silico studies.”

Respond: The “in vitro” and “in silico” has been italic (Line 358).
